# General Construction of Amine via Reduction of N=*X* (*X* = C, O, H) Bonds Mediated by Supported Nickel Boride Nanoclusters

**DOI:** 10.3390/ijms23169337

**Published:** 2022-08-19

**Authors:** Da Ke, Shaodong Zhou

**Affiliations:** 1Zhejiang Provincial Key Laboratory of Advanced Chemical Engineering Manufacture Technology, College of Chemical and Biological Engineering, Zhejiang University, Hangzhou 310027, China; 2Institute of Zhejiang University—Quzhou, Zheda Rd. #99, Quzhou 324000, China

**Keywords:** nickel boride, primary amine, hydrogenation, reductive amination

## Abstract

Amines play an important role in synthesizing drugs, pesticides, dyes, etc. Herein, we report on an efficient catalyst for the general construction of amine mediated by nickel boride nanoclusters supported by a TS-1 molecular sieve. Efficient production of amines was achieved via catalytic hydrogenation of N=*X* (*X* = C, O, H) bonds. In addition, the catalyst maintains excellent performance upon recycling. Compared with the previous reports, the high activity, simple preparation and reusability of the Ni-B catalyst in this work make it promising for industrial application in the production of amines.

## 1. Introduction

Amine constitutes a vital class of chemicals abundantly existing in nature, and are widely used in industry to produce pharmaceutical drugs, agrochemicals, fine chemicals, polymers, dyes, perfumes, pigments, etc. [1,2,3,4,5,6]. In recent years, great efforts have been conducted on the synthesis of primary amines. At present, primary amines can be prepared via direct amination of alcohols [7,8], reductive amination of aldehydes or ketone compounds [9,10,11], amination of carboxylic acids [12,13], and reduction of nitriles [14,15,16,17,18], nitro compounds [19,20,21], or amides [22]. Among these methods, the reduction of N=*X* (*X* = C, O, H) bonds plays a key role. Generally, nitriles, nitro compounds and amides can be reduced to primary amines using borane [21,23,24], silane [25], hydrides [26], formats [20,27], alcohols [28], or molecular hydrogen [29]. Since Raney Ni was first prepared in 1905, it has become one of the most important catalysts for reduction. Though Raney Ni is indeed active, it suffers from high inflammability [30]. To improve this, researchers have developed a variety of homogeneous or heterogeneous catalysts. For example, non-precious metals, such as iron [31,32,33,34,35,36], cobalt [37,38,39,40,41,42,43,44,45,46,47], copper [48,49], nickel [10,11,21,24,50,51,52,53], manganese [6,54,55], and noble metals, such as palladium [19,56,57,58], platinum [59], ruthenium [8,60,61,62], rhodium [28,63,64,65], samarium [66], and iridium [67], have been employed to construct hydrogenation catalysts.

Efficient, stable, and economical hydrogenation catalysts to synthesize primary amines continue to be demanding in both academia and industry. Amorphous nickel boride is well known for its short-range ordered and long-range disordered structures, as well as their activity in liquid phase hydrogenation [68]. Li et al. [69] used Ni-B/SiO_2_ as a catalyst to reduce adiponitrile with good selectivity and a low TOF of 1.2 (Figure 1). At present, there exists only a few reports on the reduction of unsaturated bonds mediated by nickel boride [70,71]. Additionally, the unique pore structure, large specific surface area and excellent hydrothermal stability of titanium silicalite molecular sieves make them widely used in the chemical industry, environmental protection and energy conversion [72,73,74]. The diffusion path length and the aforementioned characteristics enable titanium silicalite (TS-1) molecular sieves to perform strongly as catalysts. Herein, we report on a nickel boride catalyst with TS-1 as support, for the reduction of N=*X* (*X* = C, O, H) bonds to amines with high efficiency and universality (Figure 1).

## 2. Results and Discussion

### 2.1. Catalyst Evaluation

We first examined the performances of the catalysts prepared under different conditions, including temperature, pressure, additive and solvent. More details are listed in Appendix A. Three reactions were selected to evaluate the catalysts, i.e., the hydrogenation of benzonitrile, nitrobenzene, and the reductive amination of benzaldehyde. The detailed results are shown in Table 1, Table 2 and Table 3.

During the reduction of benzonitrile, the imine intermediate would react with primary amine to generate N-benzylidenebenzylamine (B) and further hydrogenated to dibenzylamine (C). Generally, excessive ammonia can inhibit the side reaction with the primary amine [75]. Moreover, acetylation reactions, using highly acidic or basic additives, can also promote the selectivity of primary amines [76,77,78]. In the model reaction, ammonia was not added to the reaction system in order to evaluate the intrinsic performance of the catalysts. Surprisingly, highly selective generation of primary amines was facilitated. It turned out that both the preparation temperature and the Ni content affect the performance of the catalyst: lower temperature favors a high activity of the catalyst, while a Ni content ~12% is optimal for the catalytic efficiency.

Methanol, ethanol, isopropanol and toluene were tested as the solvent (Appendix A), and isopropanol outperformed the others.

The reaction temperature and hydrogen pressure were simply screened (Appendix A), and 120 °C and 4.0 MPa were shown to be optimal.

Next, as shown in Table 2, a longer time was required to convert nitrobenzene completely, indicative for a slightly lower activity of the catalyst toward nitro reduction. Though a higher Ni content (18.6%) affords a higher TOF, the yield may not be favored.

Further, reductive amination of benzaldehyde was carried out using the nickel boride catalysts. To promote the selectivity of the target product, the critical point is to avoid further conversion of the product. To this end, it is necessary to use excessive ammonia to suppress the side reaction. As shown in Table 3, when the same amount of nickel was added, the TOF values did not change much.

Considering both the TOF value and the selectivity of the target product, the Ni_12.4_-30 catalyst was selected for further investigation.

### 2.2. Characterization of Ni_12.4_-30

In order to clarify the actual content of metallic Ni in the catalyst, the accurate mass content of Ni was obtained through the ICP-OES test. The theoretical nickel content in the catalyst was 12.4%, and the experimental data was 12.1%, which was the normal error range (Appendix A). Thus, there was no loss of Ni during the preparation process.

The XRD pattern of Ni_12.4_-30, shown in Figure 1, indicates that there was no obvious change on the TS-1 support after loading, implying that the loaded nickel boride component possesses an amorphous structure, in line with previous findings [70]. The rest moiety of the catalyst did not exhibit other diffraction peaks, regardless of the reduction temperature and Ni loading (Appendix A). It is worth noting that nickel boride may react with ethanol at high temperatures to form metallic nickel [79]. The characteristic diffraction peaks for metallic nickel were not found in the used catalyst, therefore, the stability of the nickel boride structure was thus justified, ruling out the possibility that metallic nickel generated in-situ serves as the active species.

In order to further identify the chemical state of the catalyst, Ni_12.4_-30 (both the fresh and the recycled ones) were subjected to XPS analysis, and the results are shown in Figure 2. The signals of high-resolution XPS spectra that emerged at around 860 and 190 eV correspond to Ni and B, respectively [80]. The peaks at 853 and 856 eV in Ni 2p_3/2_ are ascribed to the metallic nickel and oxidized nickel. The XPS spectrum of pure nickel boride alloy has only one peak of Ni(0), while the peak of Ni(II) appears when nickel boride is supported, in line with previous reports [69,70,71,81]. The peaks at 188 and 192 eV in B 1s are assigned to elemental and oxidized boron, respectively. The peaks of pure boron in B 1s at 187 eV (< 188 eV) may result from Ni-B interaction. No significant difference in chemical states of Ni and B appears in the used catalyst, indicative of the catalyst’s high stability.

The morphologies of the Ni_12.4_-30 catalyst was investigated using TEM. As shown in Figure 3a, the nickel boride species correspond to nanoparticles ranging 10~40 nm diameter with a mean size of 17 nm. A smaller particle size indicates a higher surface energy, and the diameter 17 nm is much smaller than that of pure nickel boride alloy (60 nm) [82], which benefits from the porous structure of TS-1. Most likely, the high activity of this catalyst generates these results. The SAED was employed to determine the crystal structure of nickel boride. There are halo diffraction rings rather than distinct dots in the SAED image, confirming the amorphous structure of nickel boride, in good agreement with XRD patterns. The EDS revealed that the nickel boride comprised of Ni (60%) and B (40%), similar to Ni_2_B.

According to the characteristic results, the high activity of the Ni_12.4_-30 catalyst may result from three aspects. First, the amorphous nickel boride possesses a large number of coordinatively unsaturated active centers on the surface, and a higher surface energy is conducive to the adsorption and conversion of reactants. Second, the electron-transfer from Ni to B causes polarization of the active center and is thus beneficial for Lewis interactions with the reactants. Third, suitable Ni loading dispersed on TS-1 promotes a proper particle size and prevents aggregation, crystallization, and deactivation.

### 2.3. The Reduction of Nitrile

In order to test the universality of the selected catalyst (Ni_12.4_-30), the reduction of various nitriles were carried out under optimal conditions. Ammonia was added into the reaction system to avoid side reactions. Consequently, for most aromatic nitriles, ideal conversion (100%) and primary amines yield (>90%) were obtained (Table 4, entries 1–14). However, when picolinonitrile or 2-aminobenzonitrile were the substrate, a much lower rate of conversion occurred. By contrast, when aliphatic nitriles were subjected to the same conditions, the reaction proceeded very inefficiently (Table 4, entries 15–16, 18–20). It was interesting to note that although the performance of adiponitrile, cyclohexanecarbonitrile and butyronitrile were poor, the performance of dodeconitrile was exceptionally good. This abnormal phenomenon might be ascribed the long carbon chain of dodeconitrile.

### 2.4. The Reduction of Nitro Compounds

Further, the catalyst was tested with the hydrogenation of aromatic nitro compounds to primary amines. Under the same conditions to nitrile reduction, more time was needed to convert nitro to amino (see Table 5). In spite of the relatively lower activity toward nitro reduction, the catalyst mediates selective generation of primary amines. The substitution groups on the phenyl ring do not have much effect on the reduction process.

### 2.5. The Reduction for Aldehyde and Ammonia

Further, the catalytic performance of Ni_12.4_-30 toward reductive amination of aldehyde was examined. Various aldehydes were employed, and the amination results are shown in Table 6. In general, all selected carbonyl compounds were converted to the corresponding amines with excellent yields upon reductive amination. As compared to aromatic aldehydes, aliphatic substrates are relatively less reactive, thus a slightly longer time is required for them to be completely converted.

In order to test the reusability of the catalyst, the Ni_12.4_-30 species was used fifteen times consecutively with 10 mmol scale. The conversion of benzonitrile, nitrobenzene, and benzaldehyde to amines were all tested. Surprisingly, no obvious loss of activity was observed (Figure 4). Furthermore, the catalytic performance of Ni_12.4_-30 was compared with the commercial Raney Ni (Table 7, see more details in Appendix A). For the reduction of benzonitrile, Ni_12.4-30_ catalyst exhibits higher selectivity of benzylamine under ammonia-free conditions. For the other two reactions, there was no obvious difference between Ni_12.4_-30 and Raney Ni.

## 3. Experimental

### 3.1. Catalyst Preparation

A typical process for catalyst preparation was followed. NiCl·6H_2_O was dissolve in 50 mL deionized water, the TS-1 molecular sieve was added, and this was stirred for 0.5 h at 30 °C. After that, 1 M NaBH_4_ solution was added to the suspension while stirring, and stirring continued for 2 h. Finally, the suspension was filtered and washed to obtain a solid catalyst, which was then subjected to vacuum drying at 50 °C for 2 h. The catalyst is named Ni_w_-T, in which “w” and “T” represent the mass content of nickel (compared to TS-1) and the temperature for catalyst preparation, respectively (see more details in the Support Information).

### 3.2. Catalyst Characterization

ICP data was obtained from Agilent-ICPOES730 (Santa Clara, CA, USA). The X-ray diffraction (XRD) patterns were measured at room temperature using D/max-rA with Cu-Kα radiation generated at 10 mA and 40 kV. The X-ray photoelectron spectroscopy (XPS) analysis was carried out by using Thermo Scientific K-Alpha (Waltham, MA, USA) with Al-Kα radiation. The morphological information was measured by a transmission electron microscope (TEM) conducted using a Thermo Scientific Talos F200S coupled with X-ray spectroscopy (EDS).

### 3.3. Catalyst Activity Measurement

Nitrile, catalyst and solvent were mixed in a 100 mL volume autoclave equipped with PTFE and magnetic pellet. The kettle was filled with 0.5 MPa ammonia gas and heated to 120 °C; at this temperature, 4.0 MPa H_2_ was pressed in, and then reaction was started. During the process, the system pressure was controlled between 4.0 ± 0.1 MPa. After reaction (under constant pressure), the autoclave was cooled and degassed, the reaction solution was filtered to recover the catalyst, and the filtrate was concentrated to determine the conversion by GC. ^1^H NMR and ^13^C NMR spectrum data were recorded by a Bruker DRX-400 spectrometer (Billerica, MA, USA) using CDCl_3_ or DMSO-d_6_ as solvent at 298 K. Gas chromatography (GC) was performed on Agilent chromatography with a SE54 column. More details in support information (for spectra, see Appendix A).

## 4. Conclusions

In conclusion, we have presented a nanostructured nickel boride catalyst than can be used for the efficient reduction of nitrile, nitro compounds, and imine groups. This catalyst was prepared via chemical reduction at room temperature with an average particle size of 17 nm and homogeneous distribution. XRD and SAED justified the amorphous structure of nickel boride. The Ni_12.4_-30 catalyst has been proven to be highly active towards all three reactions, with high TOF values. Furthermore, recycling tests proved that the catalysts are robust for consecutive use. In addition, the performance of the Ni_12.4_-30 catalyst is comparable to commercial Raney Ni, but it is safer for storage. The promising prospect of the nickel boride catalyst for industrial application has thus been proven.

## Data Availability

Additional figures are available in the Appendix A.

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
