# Peer review of "General Construction of Amine via Reduction of N=X (X = C, O, H) Bonds Mediated by Supported Nickel Boride Nanoclusters"

_ijms, 2022, doi:10.3390/ijms23169337_

Round 1

Reviewer 1 Report

The authors synthesized a series of materials based on nickel boride and studied their activity in key reactions for the production of amines: catalytic reduction of nitriles (hydrogenation), nitroarenes (hydrogenation), and aldehydes (reductive amination). Рreliminary tests were conducted, a leader was selected, its activity was demonstrated on a wide range of substrates, stability and the ability to maintain activity after repeated recycling were also very good. The article describes interesting results and may be published after revision.

1) Perhaps the introduction should mention catalytic amination approaches using alternative reducing agents such as CH (ACS Catal. 2022, 12, 12, 7142–7198).

2) "via" should be written in italics (line 27 and further in the text)

3) In Scheme 1, hydrogen (H2) should be added above the arrows.

4) the authors carried out experiments with a loading of 5 mmol, which corresponds to approximately 500 mg of the mass of the product. I'm not sure if it's acceptable to write yields and conversions to tenths of a percent with such loading (see Synlett 2010, No. 18, 2701–2707). In my opinion, only integer values ​​should be entered in tables (23 instead of 23.4, and so on). To a greater extent, this concerns the conversion determined by GC.

5) In tables 1-3, in the top diagram, write "cat." above the arrow, and also indicate the pressure of hydrogen (as in table 1) and ammonia (in table 3), so that the schemes are uniform. In the footnotes of Tables 1-3, in (a) replace "50 mg Ni12.4-30 catalyst (about 2.0 mol% Ni)" with "catalyst" because the catalyst and its loading varies.

6) The authors conducted a thorough study of the structure of the Ni12.4-30 (leader catalyst), however, the conclusion about the relationship between its structure and activity (lines 132-138) is not correct without comparison with similar parameters of other synthesized catalysts. If there is such data (or at least some of them), they should be included in the supporting information.

7) Tables 4-6 showing the scope turned out to be too large, which makes them difficult to read. They should be made more compact. Perhaps they should be reduced by replacing the structures with columns with substituents (er) or by representing them as a reaction scheme with the products displayed below it and indicating the yield, conversion and time next to each structure.

8) The figure S5 should be in the main text, it is important.

9) Table S3 should also be mentioned in the text. It would be very good if the authors compared the obtained TON and TOF with heterogeneous analogues from recent publications.

10) the format of the references should be checked, perhaps there are errors.

Author Response

Dear Editor,

Many thanks for the information concerning our manuscript submitted to International Journal of Molecular Science. The manuscript has been revised according to the comments from the reviewers, and the details are as follows:

Reviewer 1:

  1. Perhaps the introduction should mention catalytic amination approaches using alternative reducing agents such as CH (ACS Catal. 2022, 12, 12, 7142–7198).

Reply: Thanks for the suggestion. We have cited the reference (ACS Catal. 2022, 12, 12, 7142–7198) in the manuscript as reference 28 and added proper comments.

  1. "via" should be written in italics (line 27 and further in the text)”;”

Reply: Thanks for your careful check. We have corrected them in the manuscript.

  1. In Scheme 1, hydrogen (H2) should be added above the arrows.”

Reply: Thanks for your suggestion. we have redrawn Scheme 1 carefully with more details, and compared the difference between this work and the reported literature.

  1. The authors carried out experiments with a loading of 5 mmol, which corresponds to approximately 500 mg of the mass of the product. I'm not sure if it's acceptable to write yields and conversions to tenths of a percent with such loading (see Synlett 2010, No. 18, 2701–2707). In my opinion, only integer values ​​should be entered in tables (23 instead of 23.4, and so on). To a greater extent, this concerns the conversion determined by GC.

Reply: Many thanks for your comments. Indeed, under the employed loading of this manuscript, only integer value of yield may be meaningful. Thus, we have corrected yield numbers to integers in Tables 1~6, as well as the ones in support information.

  1. In tables 1-3, in the top diagram, write "cat." above the arrow, and also indicate the pressure of hydrogen (as in table 1) and ammonia (in table 3), so that the schemes are uniform. In the footnotes of Tables 1-3, in (a) replace "50 mg Ni12.4-30 catalyst (about 2.0 mol% Ni)" with "catalyst" because the catalyst and its loading varies.

Reply: Thanks for your suggestion. We have revised the manuscript accordingly.

  1. The authors conducted a thorough study of the structure of the Ni12.4-30 (leader catalyst), however, the conclusion about the relationship between its structure and activity (lines 132-138) is not correct without comparison with similar parameters of other synthesized catalysts. If there is such data (or at least some of them), they should be included in the supporting information.”

Reply: Thanks for your comments. In the text, we speculate the high activity of Ni12.4-30 results from three aspects. First, the amorphous nickel boride had a large number of coordination unsaturated active centers on the surface. The amorphous structure has more surface unsaturated centers, which has been reported previously (references 68, 80); according to the results of XRD, only the crystalline diffraction peaks of the support appears, indicative for the existence of amorphous nickel boride. It is undeniable that comparisons are needed to explain the source of the catalyst's activity. Thus, in the second and third points, we analyzed the results of XPS and TEM of Ni-30 and found that its structure corresponds exactly to that in the literature: the interaction between boron and nickel atoms led to a significant result of XPS, and the addition of support led to a smaller average particle size (reference 82, 80). Therefore, the relationship between activity and structure in the text is a speculation based on the existing literature and the results of our actual characterization.

  1. Tables 4-6 showing the scope turned out to be too large, which makes them difficult to read. They should be made more compact. Perhaps they should be reduced by replacing the structures with columns with substituents (er) or by representing them as a reaction scheme with the products displayed below it and indicating the yield, conversion and time next to each structure.

Reply: Thanks for your suggestion. We have adjusted the tables accordingly.

  1. The figure S5 should be in the main text, it is important.”

Reply: Thanks for your comments. we have moved Figure S5 into the main text as Figure 4.

  1. Table S3 should also be mentioned in the text. It would be very good if the authors compared the obtained TON and TOF with heterogeneous analogues from recent publications.”

Reply: Thanks for your suggestion. We have added Table S3 to the main text as Table 7, and kept Table S3 in the support information because of some details. On the other hand, we have compared the TOF between the reported publication and this work in Scheme 1. Since few nickel boride catalyst for primary amine formation, there is no more literature for comparison.

  1. the format of the references should be checked, perhaps there are errors.

Reply: Thanks for your suggestion. We have checked the format of all reference to avoid mistakes.

In addition, all changes were marked with track changes. We hope that the manuscript now is qualified for publication in IJMS.

Looking forward to your reply.

Thanks and best regards,

Shaodong Zhou

Reviewer 2 Report

The topic of the article fits well with the special issue: Nanoparticles for Catalysis.

At present, catalysts based on nickel borides are widely used in organic chemistry, however, the composition and properties of the catalyst vary depending on the specific preparation method. The article describes the evaluation and characterization of nickel boride catalyst with TS-1 as support and the results obtained on studies of the reduction of compounds with the N=X (X = C, O, H) bonds to amines with the use of this catalyst.

Major recommendations.

In my opinion, the main drawback of this work is the lack of sufficient information on the differences and advantages of the obtained results and obtained catalyst in comparison with the known data. The authors indicated that there are a few reports on the reduction of N=X bonds mediated by nickel boride [69-71]. It is important to indicate more clearly what is the difference and what are the advantages of new results obtained and the obtained catalyst in comparison with the known data. In my experience, in the introduction, the authors usually present two schemes, which demonstrated what has been done before and what has been done in this work (in comparison with known results). I would recommend the authors to correct the paper taking into account the above remarks and also to correct Scheme 1, which gives too little information. It is also important to provide more clearly formulated results and conclusions in Abstract and Conclusions sections and to indicate in Abstract and Conclusions advantages of new results and the obtained catalyst in comparison with the known data. The title and names of authors should be added to the supporting information.

Minor recommendations.

First of all, I would like to ask the authors to carefully check the English language in the article, because sufficient English correction is necessary. Besides, there are many minor flaws in the text. For example, "20 mL isopropanol" should be "20 mL of isopropanol" or better "isopropanol (20 mL). The same is true for other reagents. The word "nitro" is better used as "nitro compound" or "nitro group", etc. There must be a space between the word and the ref. in square brackets and between numbers and words (20 mL, 17 nm, 50 mg instead of "20mL, 17nm, 50mg", etc.).

Given the practical usefulness of the work, the results of which can potentially be used not only for laboratory research, but also for possible applications on a wider scale, the results obtained deserve to be published. I suggest the major revision for the manuscript before it will be accepted.

Author Response

Dear Editor,

Many thanks for the information concerning our manuscript submitted to International Journal of Molecular Science. The manuscript has been revised according to the comments from the reviewers, and the details are as follows:

Reviewer 2:

  1. In my opinion, the main drawback of this work is the lack of sufficient information on the differences and advantages of the obtained results and obtained catalyst in comparison with the known data. The authors indicated that there are a few reports on the reduction of N=X bonds mediated by nickel boride [69-71]. It is important to indicate more clearly what is the difference and what are the advantages of new results obtained and the obtained catalyst in comparison with the known data. In my experience, in the introduction, the authors usually present two schemes, which demonstrated what has been done before and what has been done in this work (in comparison with known results). I would recommend the authors to correct the paper taking into account the above remarks and also to correct Scheme 1, which gives too little information. It is also important to provide more clearly formulated results and conclusions in Abstract and Conclusions sections and to indicate in Abstract and Conclusions advantages of new results and the obtained catalyst in comparison with the known data. The title and names of authors should be added to the supporting information.”

Reply: Many thanks for the comments. We have reviewed and described the literature in the Abstract, Introduction and Conclusion sections. For nickel boride, there are indeed few literatures that using it for generation of primary amines. Reference 69 used nickel boride supported on SiO­2 to reduce adiponitrile for hexamethylenediamine by Gas-phase reduction. On the other hand, references 70 and 71 mainly use nickel boride catalysts to reduce carbon-carbon double bonds and reduce carbonyl groups to generate alcohols, and they had a relatively narrow scope of application. We further revised scheme 1 and added more details. The comparison of this work with that of reference 69 and proper comments were added. In addition, the title and names of authors were added to the supporting information.

  1. First of all, I would like to ask the authors to carefully check the English language in the article, because sufficient English correction is necessary. Besides, there are many minor flaws in the text. For example, "20 mL isopropanol" should be "20 mL of isopropanol" or better "isopropanol (20 mL). The same is true for other reagents. The word "nitro" is better used as "nitro compound" or "nitro group", etc. There must be a space between the word and the ref. in square brackets and between numbers and words (20 mL, 17 nm, 50 mg instead of "20mL, 17nm, 50mg", etc.).

Reply: Many thanks for your suggestion and corrections. We have made corrections accordingly, and checked entire text carefully to avoid syntax errors.

In addition, all changes were marked with track changes. We hope that the manuscript now is qualified for publication in IJMS.

Looking forward to your reply.

Thanks and best regards,

Shaodong Zhou

Round 2

Reviewer 2 Report

The authors corrected the article in accordance with the comments and it has become much better. Although there are some shortcomings in the text (e.g., there must be a space between the word and the ref. in square bracket), I hope they will be corrected by the technical editor and recommend the article for publication.